# Impact of *OsBadh2* Mutations on Salt Stress Response in Rice

**DOI:** 10.3390/plants11212829

**Published:** 2022-10-24

**Authors:** Zakaria H. Prodhan, Shah A. Islam, Mohammad S. Alam, Shan Li, Meng Jiang, Yuanyuan Tan, Qingyao Shu

**Affiliations:** 1National Key Laboratory of Rice Biology, The Advanced Seed Institute, Zhejiang University, Hangzhou 310058, China; 2College of Life Sciences, Neijiang Normal University, Neijiang 641100, China; 3Agronomy Division, Bangladesh Rice Research Institute, Gazipur 1701, Bangladesh

**Keywords:** aroma, *OsBadh2*, CRISPR/Cas9, salinity, biochemical analysis, rice

## Abstract

Mutations in the *Betaine aldehyde dehydrogenase 2* (*OsBadh2*) gene resulted in aroma, which is a highly preferred grain quality attribute in rice. However, research on naturally occurring aromatic rice has revealed ambiguity and controversy regarding aroma emission, stress tolerance, and response to salinity. In this study, mutant lines of two non-aromatic varieties, Huaidao#5 (WT_HD) and Jiahua#1 (WT_JH), were generated by targeted mutagenesis of *OsBadh2* using CRISPR/Cas9 technology. The mutant lines of both varieties became aromatic; however, WT_HD mutants exhibited an improved tolerance, while those of WT_JH showed a reduced tolerance to salt stress. To gain insight into the molecular mechanism leading to the opposite effects, comparative analyses of the physiological activities and expressions of aroma- and salinity-related genes were investigated. The WT_HD mutants had a lower mean increment rate of malondialdehyde, superoxide dismutase, glutamate, and proline content, with a higher mean increment rate of γ-aminobutyric acid, hydrogen peroxide, and catalase than the WT_JH mutants. Fluctuations were also detected in the salinity-related gene expression. Thus, the response mechanism of *OsBadh2* mutants is complicated where the genetic makeup of the rice variety and interactions of several genes are involved, which requires more in-depth research to explore the possibility of producing highly tolerant aromatic rice genotypes.

## 1. Introduction

Rice is the most important cereal crop, considered as a staple food, and it has great variations in grain quality, which determine its market value and consumer acceptance all over the world [1]. A key gene, *Betaine aldehyde dehydrogenase 2* (*LOC_Os08g0424500*; also known as *FGR*/*fgr* or the *Badh2*/*badh2* or *OsBadh2*/*osbadh2* gene), which is related to the synthesis of 2-acetyl-1-pyrroline (2AP), an aroma component in rice, determines whether the rice grain is aromatic or non-aromatic [2,3,4]. Usually, aromatic rice cultivars are lower yielding, have inferior agronomic traits, and are more susceptible to environmental stress than non-aromatic varieties [5]. Some researchers observed that osmotic stress (i.e., drought and salinity) has a favorable effect on aroma quality [6] and produces a high quantity of proline through the intermediary of Δ^1^-pyrroline-5-carboxylic acid synthesis [7]. Experiments on rice callus and rice plants revealed that proline plays an important role in plant responses to water deficits and salt stressors [8,9]. Several other potential metabolites and enzymes, such as free amino acids, glutamate, polyamines (PAs), and γ-aminobutyric acid (GABA), have been found to actively respond to environmental stress and are also believed to be responsible for the high-quality aroma synthesis in aromatic rice [10]. GABA could be synthesized by the polyamine degradation pathway (in which glutamate is converted to γ-aminobutyraldehyde “GABA-ald", which is then converted to GABA via the activation of a functional *OsBadh2* gene) or through the GABA shunt pathway (where glutamate is directly converted to GABA by the activity of glutamate decarboxylase “*OsGAD*” genes) [11]. The nature and level of stress, as perceived by cellular changes in Ca^2+^ and/or H^+^ concentration, activate *OsGAD* genes to synthesize GABA through the GABA shunt pathway, which passively modulates the aroma component (2AP concentration) by converting glutamate to GABA [11,12]. When GABA binds to a GABA-like receptor, Ca^2+^ is released from intracellular storage, increasing the amount of Ca^2+^ in the cytosol. The cytosolic Ca^2+^ increases the Calcium/Calmodulin (Ca^2+^/CaM) complex, the stress response signal, and activates the genes involved in the stress response [11,12,13].

The accumulation of common osmolytes, such as reducing sugars and free amino acids (i.e., cysteine, arginine, and methionine); the activity of radical scavenging enzymes, including superoxide dismutase (SOD), catalase (CAT), ascorbate peroxidase (APX), and glutathione peroxidase (GPX); the level of aroma were all shown to be influenced by salt treatment [14]. During salinity stress, the salt-resistant rice cultivar Pokkali was found to have higher levels of antioxidants, such as ascorbate (ASC) and glutathione (GSH), and higher activity of ROS scavenging enzymes, such as catalase (CAT), than the salt-sensitive rice cultivar Pusa Basmati [10,14]. However, the mechanism by which these common metabolites are channeled into the aroma or salt tolerance pathways in salt-stressed aromatic rice cultivars has not been fully examined; only that the response varies by variety has been addressed [14,15].

Research on aromatic rice, aroma genes, and salt stress shows different hypotheses and speculates on various ambiguities in the response of the aroma gene (*OsBadh2*) to environmental stressors. It was shown that *OsBadh2* does not play a role in abiotic stress tolerance, whereas *OsBadh1* may be implicated in the salt stress response in rice [16]. There is a correlation between the aromatic rice types and a BADH1 protein haplotype (PH2) with the alterations lysine_144_ for asparagine_144_ and lysine_345_ for glutamine_345_ due to the presence of single-nucleotide polymorphisms (SNPs) [17]. The simultaneous increase in the salt concentration and *OsBadh1* gene transcript in the leaf tissue, along with a significant difference in the ability to produce mature seed in both non-aromatic and aromatic rice in the presence of salt, indicated the involvement of both the *OsBadh1* and *OsBadh2* genes in salinity tolerance [11,18]. However, suppression of *OsBadh2* in non-aromatic rice produced aroma, but the transgenic plants were less tolerant to salt stress [19]. Furthermore, salt stress treatments in several aromatic rice varieties revealed considerable genotypic heterogeneity as well as significant diversity in morphophysiological traits [20]. These results give extremely strong evidence that *OsBadh2* also contributes to salt tolerance in rice. Due to the fact that aromatic rice varieties have a lack of the functional *OsBadh2* gene, it looks probable that aromatic rice varieties will have a lower tolerance to salt than non-aromatic rice types. So far, there are only a very few scientific publications available with partial investigations into the relationship between aroma and salt resistance in cultivated rice types, which might contribute to the ambiguity of this critical scientific question.

There have been several studies on transcription factors (TFs) from various families being activated in stress-response pathways, which are crucial for integrating salt sensory pathways and tolerance responses [13,21]. Genes belonging to the transcription factor family are differentially expressed in response to elevated external salinity including basic leucine zipper (bZIP) [22], WRKY [23], APETALA2/ETHYLENE RESPONSE FACTOR (AP2/ERF) [24], MYB [25], basic helix-loophelix (bHLH) [26], and NAC [27] families. The presence of NAC proteins (plant-specific TFs) confers salt stress resistance in plants [13,28,29]. Several stress-inducible genes, including *OsNAC2*, *OsNAC3*, *OsNAC5*, *OsNAC022*, and *OsNAC106*, were shown to be upregulated in transgenic plants overexpressing NAC proteins [29,30]. Plant MYB-type TFs have a role in plant development as well as stress response, particularly salt stress [31,32]. Certain MYB TFs may also affect the expression of certain transporter genes. For example, *OsMYBc* binds to the AAANATNY motif in the promoter of *OsHKT1;1*, enhancing its expression [33]. The *OsDREB1C* gene belongs to the DREB family and the *OsERF068* gene to the ERF family, both of which are members of the AP2/ERF family, which is a wide family of TFs found in plants that play an essential role in salt stress tolerance mechanisms [34,35,36]. In addition, it was shown that the genes encoding high-affinity K^+^ transporters (HAK) *OsHAK1*, *OsHAK5*, *OsHAK16*, and *OsHAK21* play critical roles in K^+^ homeostasis under stressful situations, but their expression patterns vary [37,38,39].

The creation of precise mutations in the *OsBadh2* gene and the comparison of the responses of wild-type and mutant lines to stress conditions could provide valuable information on the mechanism of regulating physico-biochemical compounds, scavenging enzymes, and molecular responses of aroma genes to stress. Hence, the present experiment was designed to introduce mutations into the *OsBadh2* genes of two rice cultivars using CRISPR/Cas9 and to expose them to salt stress at the seedling stage. Free amino acids, ROS-associated chemicals, antioxidant-related enzymes, and the expression of several genes linked to aroma and salinity were assessed. The mutant lines of both varieties became aromatic, but those of WT_HD (i.e., HDM1 and HDM2) showed higher resistance to salt stress than those of WT_JH (i.e., JHM1 and JHM2). Malondialdehyde, superoxide dismutase, glutamate, and proline increased more slowly in HD than in JH mutants. On the other hand, the HD mutants showed a higher mean increase rate of γ-aminobutyric acid, hydrogen peroxide, and catalase than the JH mutants, and this trend was also observed in salinity-related gene expression. These phenomena indicate diverse adaptation strategies utilized by the *OsBadh2* mutant lines under salt stress. The present scientific investigation provides unique insight into the relationship between the aroma genes and salinity stress tolerance, shedding light on the prospect of creating more stress-tolerant aromatic rice lines from non-aromatic genetic backgrounds.

## 2. Results

### 2.1. Mutations of the OsBadh2 Gene and the Development of Homozygous Transgene-Free Mutant Lines

In this research, a number of transgenic plants were obtained, which were generated by designing the sgRNA targeting the 7th exon of the *OsBadh2* gene. The mutations of the T_0_ plants were confirmed by sequencing using the primer flanking the sgRNA (Appendix A), and all of the mutations were found to be small insertions or deletions (Figure 1).

The mutations with 1 bp deletions (i.e., T/C bp deletion) were found to have the same amino acid deletion with an identical protein translation or protein structure, while mutations with insertions and deletions of the A bp and TC bp were distinct protein structures in the protein model prediction [40]. Furthermore, it was observed that mutants with a 1 bp deletion were the most prevalent mutation type in the present CRISPR/Cas9 genome editing system (Figure 1).

The seeds of the T_0_ plants with mutations were harvested and grown into T_1_ plants, which were further tested for both mutations and the presence of T-DNA. The seeds from mutated, homozygous T_1_ plants were harvested and grown into T_2_ lines.

### 2.2. Plant Phenotypic Changes under Salt Stress

The International Rice Research Institute (IRRI) recommended conventional salinity stress tolerance screening concentration (i.e., 100 mM NaCl) [41] was found to be efficient for demonstrating the phenotypic differentiation between the wild-type and mutant lines. The wild-type plants showed tolerance to salt stress, but the Huaidao#5 mutant lines (i.e., HDM1 and HDM2) showed improved tolerance, while the Jiahua#1 mutant lines (i.e., JHM1 and JHM2) exhibited a reduced tolerance phenotype to salinity stress (Figure 2).

Based on the phenotypic changes, such as leaf rolling and wilting, under salinity stress, the mutant lines could be characterized as improved or highly tolerant mutants (score 1, HDM1 and HDM2) and reduced or moderately tolerant mutants (score 5, JHM1 and JHM2) with very few phenotypic changes between them. These results indicate that the wild types and their mutant lines show a varying reaction towards salinity stress depending on the rice cultivars and mutation types. The response differences of rice plants to the morphological indices were estimated and displayed in Figure 3.

In this experiment, root length (Figure 3A), shoot length (Figure 3B), root-to-shoot ratio (Figure 3C), plant fresh weight (Figure 3D), and plant dry weight (Figure 3E) were reduced in all experimental samples under saline conditions. A higher reduction rate (20.23%) in the root length was observed in JHM1, and a lower reduction rate (4.60%) was found in HDM2. The mean reduction rate of the root length was lower (9.83%) in the HD mutants (i.e., HDM1 and HDM2) than in the JH mutants (i.e., JHM1 and JHM2) (19.41%). A variation in the reduction of shoot length (11.43–15.21%), fresh weight (17.36–58.91%), and dry weight (29.74–50.44%) was observed under salinity, but the reduction rate was not uniform. However, a higher mean reduction rate of the root-to-shoot ratio (−7.25%) was observed in the JH mutants than in the HD mutants (+5.52%), which resembles the phenotypic expression of the plants.

### 2.3. Free Amino Acid Content, Aroma, and Aroma-Related Gene Expression Analysis

In this experiment, there was an increase in the concentration of stress-related free amino acids (FAAs) in all of the experimental lines (i.e., wild types and mutants) under salt treatment. The glutamate concentration increased (8.06–93.93%) under salinity, and the wild types produced more glutamate than their mutants, while the increment was highest in JHM2 (93.93%) (Figure 4A). Similarly, the proline concentration increased significantly in all of the experimental salt-treated lines, and the increment was higher in the JHM2 mutants (3.57 µmol/g FW) than in the HDM2 mutants (1.60 µmol/g FW) (Figure 4B). A lower mean increment rate of glutamate and proline content (31.29 and 950.41%, respectively) was observed in the HD mutants than in the JH mutants (72.86 and 2280.42%, respectively). Conversely, a higher mean increment rate for γ-aminobutyric acid was detected in the HD mutants (71.13%) than in the JH mutants (43.37%) (Figure 4C). These results indicate that the free amino acid content increased under salinity stress, but the increment depended on the genotypic condition of the plants.

The grain aroma score increased in the mutant lines compared to their wild types (Figure 4E), indicating that the *osbadh2* mutations are responsible for aroma in the mutant lines. The leaf aroma was induced more in mutants (7.16 to 41.65%) than in their wild types at both normal and salinity conditions (Figure 4D), pointing out that salinity has a positive influence on aroma emission. The expression of *OsBadh1* was upregulated under salt treatment for both the wild-type and mutant lines in both experimental cultivars (Figure 4F). The mean *OsBadh1* expression rate was higher in the JH mutants (50.16%) than in the HD mutants (12.14%). Salinity stress induced the expression of the *OsP5CS1* gene (Figure 4G), but the increment was not significantly different in both cultivars and in their mutants. The *OsP5CS* showed a minor reduction (1.16–3.32%) in HD mutants but a great increase in the JH mutants (82.22%) (Figure 4H). During salt stress, *OsGAD1*, *OsGAD2*, and *OsGAD4* presented low increments (78.83, 168.81, and 80.78%, respectively) in the HD mutants, while they exhibited high increments (168.64, 287.46, and 146.14%, respectively) in the JH mutants (Figure 4I,J,L, respectively). The expression of *OsGAD3* (Figure 4K) was found to be inverse in both experimental cultivars. However, the mean *OsGAD3* expression rate increased in the HD mutants (+216.38%) but decreased in the JH mutants (−64.27%). These results point out that the glutamate decarboxylase (GAD) genes, which convert L-glutamate to GABA directly by the GABA shunt pathway, responded differently in plants under salinity stress.

### 2.4. The Change in the Lipid Peroxidation, ROS Scavenging, and Antioxidant Enzyme Content Estimation

Salt stress increased the synthesis of lipid peroxidation (MDA content) and ROS nonradical (molecular) form of H_2_O_2_, both of which are extremely reactive, cytotoxic, and could destroy cell structures. The MDA content (Figure 5A) and H_2_O_2_ accumulation (Figure 5B) were significantly increased by 3.32–15.29 times and 0.59–5.64 fold, respectively, after salinity treatment. The increment was higher in the JH mutants (JHM1 and JHM2; 12.05 and 2.85 µmol/g FW, respectively) compared to the HD mutants (HDM1 and HDM2; 3.32 and 2.11 µmol/g FW, respectively). The mean increment rate of MDA was lower (123.38%) in HD mutants than in the JH mutants (153.88%), while the mean increment rate of hydrogen peroxide was higher in the HD mutants (180.06%) than in the JH mutants (150.34%). The results also demonstrated that the production of lipid peroxidation and ROS nonradical enzymes depends on the salinity response of mutants.

The antioxidant enzymes (i.e., SOD, POD (peroxidase), and CAT) were shown to be more active in all of the experimental lines when exposed to salinity stress. The highest increase in SOD activity (approximately three-fold) (Figure 5C) was detected in JHM1, and the highest increase in POD activity (approximately five-fold) (Figure 5D) was reported in the Huaidao#5 wild type, although the CAT activity was at its peak (approximately five-fold) in HDM1 (Figure 5E). These results indicate that the SOD increment (36.11–236.92%) was not uniform, the POD increment (14.50–66.66 U/mg FW) was reduced in the mutants more than in their respective WT, and the CAT increment (20–348.15%) was higher in the mutant lines than in their respective WT. However, a lower mean increment rate for the SOD content (44.62%) was observed in the HD mutants than in the JH mutants (141.20%). Conversely, a higher mean increment rate of CAT was detected in the HD mutants (197.88%) than in the JH mutants (77.88%). On the other hand, a reduced APX enzyme activity (0.47–4.83 U/g FW) was found under salinity, with the greatest reduction reported in the wild-type lines (Figure 5F). The mean reduction rate was lower in the HD mutants (−6.65%) than in the JH mutants (−13.61%).

### 2.5. Salinity-Related Gene Expression Analysis

In both cultivars, the relative expression of some salinity-related genes showed differential expressions of the mRNA transcript in all of the mutant lines compared to their wild type (Figure 6).

The differential expression of *OsNAC3* was observed in both the HD and JH mutants, while the mean downregulation rate was lower in the HD mutants (−0.92%) than in the JH mutants (−28.55%; Figure 6A). The *OsHAK5* gene demonstrated a higher mean increment rate in the HD mutants (+163.75%), while a minor mean reduction rate (−2.07%) was observed in the JH mutants (Figure 6B). The *OsDREB1C*, *OsERF086*, and *OsMYB30* genes demonstrated a higher mean increment rate in the HD mutants (663.08, 248.04, and 102.18%, respectively) than the JH mutants (454.93, 20.47, and 11.15%, respectively) (Figure 6C,D,E). An upregulation of aroma and salinity-responsive genes (*OsBadh1*, *OsP5CS1*, *OsP5CS2*, *OsGAD1*, *OsGAD2*, *OsGAD4*, *OsERF068*, *OsMYB30*, and *OsDREB1C* genes) and the differential expression of the *OsNAC3* and *OsHAK5* genes were observed under salinity, and the increment was higher in the mutants than in their WT, and the higher increment was in the HD mutants than in the JH mutants. These phenomena indicate that the salinity response mechanism is complicated, sometimes inconsistent, and dependent on the gene interaction and genetic makeup of the rice variety.

## 3. Discussion

Mutations in the *OsBadh2* gene result in aroma emission in rice. The generation of aromatic rice through genome editing by the targeted mutagenesis of *OsBadh2* has become an area of extensive research [42]. Previously, researchers successfully used CRISPR/Cas9 technologies to create precise mutations in the *OsBadh2* gene in non-aromatic rice varieties such as Zhonghua 11 [43] and ASD16 [44]. All of these studies concentrated on creating aromatic rice and quantifying the aroma component. However, the molecular mechanism of aroma expression and its interaction with environmental components have not been investigated. Moreover, it is essential to investigate the stress tolerance mechanism, ROS-scavenging process, and aroma emission pathway to analyze the feasibility of producing more tolerant aromatic rice from the non-aromatic genetic background. Thus, this research aimed to delve further into the morphological, physico-biochemical, ROS scavenging, and molecular responses of *osbadh2* mutant alleles towards salt stress in wild types and their respective CRISPR/Cas9 mutant lines.

A knockout mutation of the *OsBadh2* gene using a genome editing technique with CRISPR/cas9 technology demonstrated a successful and precise mutational event resulting in grain and leaf aroma production (Figure 4E and Figure 4D, respectively) in rice. Moreover, the mutant line demonstrated higher aroma scores compared to their wild types, indicating that the *OsBadh2* gene mutation is responsible for rice aroma. In this experiment, the most common mutations were A bp insertion and T bp deletion in the Huaidao#5 cultivar and TC bp deletion and C bp deletion in the Jiahua#1 cultivar (Figure 1). Previous studies employing CRISPR/Cas9 technology observed a similar pattern of having InDel mutations on the *OsBadh2* gene [43,44]. All of these types generated aroma, demonstrating that diverse mutational forms in *OsBadh2* may result in aroma emission in rice [45].

In the current study, the experimental lines showed diversity in the morphological indices (i.e., root length, shoot length, root-to-shoot ratio, fresh weight, and dry weight), and the morphological parameters were substantially different between the control and salt-treated lines (at a 5% level) (Figure 3). There was a distinct change in the reduction of biomass as well as morphological features in all samples under salinity, but the leaf aroma scores increased (Figure 4D). Previously, similar fluctuations in the morphological parameters in aromatic Basmati rice were identified as a result of saline stress [20]. The morphophysiological roles of the *OsBadh2* gene in a variety of salt-stressed situations were also investigated [19]. In comparison to the wild type, the seedling development rates (as measured by shoot length and weight, root length and weight, and root number) were more or less reduced in the *Osbadh2*-deficient lines at various salt concentrations (50 and 100 mM NaCl) [19]. On the other hand, the *BADH2* transcripts responded erratically to salt treatment (17, 50, and 170 mM NaCl), with no discernible trend in any of the aromatic or non-aromatic types [16]. They [16] concluded that there was no significant association between salt treatment and BADH2 transcript levels, which might account for the lack of a considerable performance loss in the aromatic rice types caused by the *OsBadh2* gene mutations. However, the present experiment indicated that the mutant lines of WT_HD (i.e., HDM1 and HDM2) exhibited improved tolerance, while those of WT_JH (i.e., JHM1 and JHM2) showed a reduced tolerance to salt stress (Figure 2), necessitating a more in-depth examination of the physico-biochemical and molecular responses of the wild-type and mutant lines to salt stress.

Salinity is a major abiotic stressor that negatively impacts rice growth, productivity, and grain quality [46]. Aromatic rice cultivars with naturally occurring spontaneous *OsBadh2* gene mutations are often more vulnerable to salt stress [47,48]. Additionally, there is a definite correlation between the participation of common metabolites (such as Pro, P5C, GABA, and PAs) in the aroma synthesis and stress alleviation of aromatic rice cultivars [15]. The accumulation of free amino acids is a frequent response of plants to environmental stress. In tomato leaves [49] and rice roots and shoots [50], a variation in the content of a frequently responsive amino acid was detected. In the current investigation, all of the targeted amino acids (i.e., glutamate, proline, and GABA) were found to be higher in the salt-treated lines than in the nontreated lines (Figure 4A,B,C, respectively). Similar increases in Glu, Pro, GABA, and arginine (Arg) were detected in both the shoot and root of the rice seedlings treated with cadmium (Cd) [50]. Proline is a critical osmolyte that plays a role in exogenous salt tolerance in a variety of crop species [51]. After salt treatment, the maximum proline accumulation (3.71 mol/g FW) was observed in the JHM2 mutant. GABA is known to protect plants from salt stress by acting as an osmo-protectant [52,53]. The GABA content of the Jiahua#1 wild-type and mutant lines (i.e., WT_JH and JHM2 lines) appeared to be higher than that of the Huaidao#5 wild-type and mutant lines. Previously, an increase in the GABA content of the non-aromatic IR-64 cultivar was reported under salt stress to compensate for osmotic imbalances [15]. In general, when the wild-type and *osbadh2* mutant lines were subjected to salt stress, there was a difference in the accumulation of free amino acids, and the amount was higher in the JH mutants compared to the highly tolerant HD mutants.

Reactive oxygen species (ROS), such as hydrogen peroxide and hydroxyl radicals, are metabolic byproducts of plant cells that contribute to lipid peroxidation (MDA content), protein denaturation, and salt-induced persistent wilting [14]. The current investigation found that salt-treated lines had greater MDA and H_2_O_2_ levels than control lines (Figure 5A,B). Previously, a similarly strong relationship between salt stress and increased lipid peroxidation was reported in rice plants [54,55]. Additionally, it was reported that the intensity of the oxidative stress differs across plant tissues, with root tissues being the most affected by salinity-induced oxidative stress, followed by mature and young leaves [56]. Total reactive oxygen species (ROS), lipid peroxidation, and electrolyte leakage (EL) were all significantly increased in rice root tissues under salt stress compared to control [56]. Another study discovered an increase in H_2_O_2_ (by 176%) and malondialdehyde (MDA, by 94%), confirming salinity (100 mM NaCl)-induced oxidative stress in tomatoes [57]. Increased levels of H_2_O_2_ (by 50%) and MDA (by 25%) were observed in maize plants exposed to salt stress (120 mM NaCl) compared to controls [58]. Thus, the degree of oxidative stress and the adaptive response to stress differed among the genotypes within a species.

A sophisticated antioxidative enzyme system, comprising SOD, POD, GPX, CAT, APX, and GR (glutathione reductase, EC 1.6.4.2), was used to eliminate ROS [14]. Salinity induces an increase in total peroxidase activity, which might be linked to changes in cell wall mechanical properties which, in turn, could be linked to salt adaptation [59]. The current experiment detected that salinity stress affected the antioxidant enzymes activity (SOD, POD, and CAT, as shown in Figure 5C,D,E, respectively), which is consistent with previous findings [60] that found that the antioxidant enzyme activity was higher in rice-tolerant varieties than susceptible varieties under salt stress conditions. The increased activity of the antioxidant enzymes CAT, POD, and SOD corresponds to an increased salinity tolerance [61]. In the current research, it was found that salinity significantly enhanced the antioxidant enzyme activity in salt-treated lines compared to nontreated control lines. However, a drop in the APX enzyme activity (Figure 5F) was observed in the current experimental materials, which is consistent with the recent findings [62] that APX activities were lower in tolerant rice lines compared to susceptible rice lines. The lowered activity of the APX enzyme was more noticeable in the high dosage (150 mM NaCl) treatment of 64 h, while in the present experiment, the concentration was 100 mM NaCl, and the treatment duration was 96 h (salt treatment). A short period of salt exposure (96 h) to 14-day old rice seedlings (highly responsive to stress) resulted in the increased synthesis of MDA and H_2_O_2_, as well as an elevated antioxidant enzyme activity, which activated the signaling pathway in a favorable way to compensate for the salinity stress. This occurrence suggests that the mutant lines employed diverse adaption strategies under the salinity stress.

Changes in the gene expression levels, or epigenetic regulation, are the most common molecular response of plants towards any environmental stimuli [63]. In this research, upregulation of *OsBadh1*, *OsP5CS1*, *OsP5CS2*, *OsGAD1*, *OsGAD2*, *OsGAD4*, *OsERF068*, *OsMYB30*, and *OsDREB1C* genes was observed during salt stress in both wild types (Huaidao#5 and Jiahua#1) and their mutant lines, which might corroborate their role in salinity stress response (Figure 4 and Figure 6). Furthermore, the upregulation of the *OsNAC3* and *OsGAD3* genes in Huaidai#5 wild-type and mutant lines (i.e., HDM1 and HDM2) in both conditions (i.e., salt-treated and nontreated), but the downregulation in the Jiahua#1 wild-type and mutant lines (i.e., JHM1 and JHM2), indicated varietal response differences to salinity stress. Previously, gene expression studies had shown that salt, drought, cold, and high light intensity increased *OsBadh1* mRNA expression [64]. Furthermore, transferring the *OsBadh1* gene from *Indica* rice (salt-tolerant) into genetically manipulated *Japonica* rice (salt-sensitive) resulted in an increased salt tolerance [65,66]. The enzyme ∆^1^-pyrroline-5-carboxylate synthetase (OsP5CS), which is involved in the production of proline, is stimulated by excessive salt, dehydration, ABA, and cold treatment [67]. Moreover, when exposed to salt stress, the expression of *OsP5CS* mRNA increased steadily in a salt-tolerant cultivar (Dee-gee-woo-gen) but only modestly in a salt-sensitive breeding line (IR28) [67]. Additionally, an increased expression of the *OsP5CS1* gene was detected under salt stress in salt-susceptible (LPT123) and salt-resistant (LPT123-TC171) rice lines [68]. Simultaneously, it was discovered that the *OsP5CS2* gene is required for salt and cold stress tolerance [69]. The *OsGAD*(*s*) genes were shown to be upregulated in conjunction with polyamine pathway gene transcripts in germinated brown rice (Shangshida No. 5) with increased GABA content [70]. Overexpression of *OsGAD* genes revealed GABA buildup in rice under short-term salinity [12]. The upregulation of multiple *GAD* genes at the mRNA level resulted in an increase in GABA levels in *tobacco* under drought stress [71].

Previously, researchers observed that the early seedling stage is highly sensitive to salt stress [72], which ultimately resulted in a reduction in the overall shoot growth and root growth [73]. These phenomena activate the key genes involved in salt stress sensing, signaling, transcriptional regulation, and genes encoding downstream functional molecules for high abscisic acid (ABA) accumulation [37]. The sophisticated mechanism underpinning salt tolerance, as well as the complexity of salt stress itself and the enormous diversity of plant responses, make the characteristic mysterious. Multiple signaling pathways could be triggered during stress exposure, resulting in comparable responses to various stimuli, revealing an overlap in the gene expression among environmental stressors [74]. Despite significant advances in our understanding of the mechanism and management of salt stress tolerance in rice, substantial gaps remain to be investigated, necessitating future research.

## 4. Materials and Methods

### 4.1. Construction of a CRISPR/Cas9 Expression Vector for OsBadh2 Gene Editing

For targeted genome editing, a CRISPR/Cas9 vector (plasmid pHUN4c12) containing the U3 RNA polymerase-III promoter [75] for expressing sgRNA and the Ubi promoter for Cas9 gene expression was utilized in this experiment. The genomic segment (approximately 1 kb) from the area of interest in the *OsBadh2* gene was chosen from the Gramene database (https://www.gramene.org/ accessed on 11 February 2018). This sequence was submitted to the CRISPR-P 2.0 program (http://cbi.hzau.edu.cn/cgi-bin/CRISPR2/ accessed on 15 February 2018), and the sgRNA oligos with the highest score were selected. The top strand was designated to be the initial strand, with a 20 bp sequence of 5’-NGG, while the bottom strand was assigned to be the opposite complement of the top strand. Finally, to match the sticky ends of the BsaI-digested plasmid, GGCA and CAAA were added to the top and bottom strands, respectively, and synthesized (Tsingke, Hangzhou, China). The two oligos were annealed in annealing buffer 5X (Beyotime Biotechnology, Shanghai, China), ligated (using T4 ligase enzyme; NEB #M0202S) to linearized pHUN4c12 (digested by BsaI, NEB), and purified using the Axygen^®^ AxyPrep DNA gel purification kit (Capitol Scientific, Austin, TX, USA). Heat shock treatment was used to transfer the ligated vector into the chemically competent *Escherichia coli* strain *DH5α*, and the insertion of the base pairing sequence was confirmed by sequencing (Tsingke, Hangzhou, China). The construct was injected into a chemically competent *Agrobacterium* (strain *EHA105*), which was then used for the transformation of rice callus.

### 4.2. Agrobacterium-Mediated Rice Transformation

Healthy mature rice seeds of the *japonica* rice varieties Huaidao#5 (WT_HD) and Jiahua#1 (WT_JH) were collected from the collaborative research center of Wuxi Hubble and Zhejiang University (Jiangsu Province, China), dehulled, and stored in a clean jar. The seeds were surface sterilized for 1 min with 70% ethanol, followed by 20 min on a shaker with a 10% (*v*/*v*) sodium hypochlorite solution containing 2% of Tween^®^20 (Ultrapure, Thermo Scientific™, Wilmington, DE, USA). After surface sterilization, the seeds were washed five times with sterilized distilled water, with the last rinse lasting 30 min. The surface sterilized seeds were placed in a Petri dish on sterilized filter paper and allowed to dry for a few minutes. The seeds were then transferred to callus induction media (2N6Y), with approximately 36 seeds per plate. The plates were sealed with parafilm and incubated at 26 ± 2 °C for 3 to 4 weeks at a 16/8 h photoperiod. The induced callus (embryogenic) was cut into tiny pieces and sub-cultured onto fresh 2N6Y medium for a further two weeks [76]. When the callus diameter reached 4 mm, it was transformed with pHUN4c12-OsBadh2 using an *Agrobacterium*-mediated transformation. The transformed calli were then cultured on a selective medium containing hygromycin and tetracycline [76]. The surviving calli were developed into plantlets in regeneration medium and incubated at 26 ± 2 °C for roughly three weeks at a 16/8 h photoperiod. The plantlets (approximately 5 cm tall) were placed in rooting media. The jar containing the rooted plants was then opened, a nutrient solution was poured on time to acclimate the plants for 3–5 days, and they were transferred to a greenhouse before being transplanted into soil and growing into T_1_ plants.

### 4.3. Molecular Characterization, Growth, and Selection of Mutants

The genomic DNA was isolated using the CTAB technique from the leaves of T_0_ transgenic rice plants. The final concentration of the extracted DNA was adjusted to 100–200 ng/L by diluting with TE buffer. In order to identify the mutation in the target area, sequencing analysis was conducted. Primers with PCR product lengths ranging from 180 to 300 bp were created online using the PrimerQuest^TM^ Tool (https://www.idtdna.com/pages/tools/primerquest accessed on 15 March 2019), and the PCR primers are listed in Appendix A. Harvested seeds (T_1_) from confirmed mutants were dried for 2–3 days in the sun. After drying, the seeds were kept at 4 °C till the next rice growing season or until further examination. To determine the presence or absence of T-DNA, leaf samples from each plant (T_1_) were collected, genomic DNA was extracted, and a primer pair (Hyg-F and Hyg-R) was used to amplify a fragment of the T-DNA in the hygromycin-resistance gene on the vector. The T-DNA-free plants with homozygous and nonsense mutations were selected for continued growth. Two distinct mutants were selected for continued development and investigation of each rice cultivar. All plants were grown to the T_2_ generation and screened for homozygous T-DNA free lines for utilization as experimental materials in this investigation.

### 4.4. Growth Conditions of the Plant Material and Salinity Stress

The rice seeds were treated for 20 min with a 2% (*w*/*v*) solution of aqueous NaClO or NaOCl, rinsed five times with deionized water, and then submerged in deionized water for one day at room temperature. After four days of germination at 28 °C, rice seedlings were transplanted into containers with standard half-modified Yoshida solution [77] and cultured in a growth chamber with a photoperiod of 14 h light (500 μmol photons m^−2^ s^−1^, 30 °C) and 10 h darkness (26 °C). For the salinity treatments, 14-day old rice seedlings were planted in half-modified Yoshida solution containing 100 μM NaCl. After 96 h of salt treatment, the rice plants were collected for phenotypic, physiological, biochemical, and molecular analysis.

### 4.5. Morphological Indices Determination

The morphological and growth responses of the 14-day old seedlings were evaluated using a procedure based on root lengths, shoot lengths, seedling fresh, and seedling dry weights [20]. The salinity tolerance status (i.e., highly tolerant, tolerant, moderately tolerant, susceptible, or highly susceptible) of the plant materials was observed and scored by following the standard evaluation score (SES) system [41,78] as illustrated in Figure 7.

The root length (cm) was measured from the root–shoot junction to the root tip. The shoot length (cm) was measured from the root–shoot junction to the top leaf tip. The ratio of the root-to-shoot length was calculated by dividing the root length by the shoot length. Immediately after harvesting, the seedling fresh weight (SFW, g) was measured to minimize the evaporation. To determine the dry weight (SDW, g), pre-weighted seedlings were covered in aluminum foil paper and kept at 65 °C for 24 h. After 24 h of drying, the seedling dry weight was determined.

### 4.6. Assessment of the Leaf and Grain Aroma

The aroma of the grain and leaves was assessed using a previously published technique [44,79,80]. Approximately 2 g of grain (dehulled) and 10 cm of leaf material (cut into little pieces) were placed on glass Petri plates. Ten milliliters of potassium hydroxide (KOH) solution was poured into each Petri plate that contained the experimental material. Immediately after the addition of alkali, the Petri plates were covered and kept at room temperature for approximately 10–30 min. After, the Petri plates were opened one by one and smelt instantly. Each sample was graded on a 1–4 scale, with 1 indicating no aroma, 2 indicating a minor aroma, 3 indicating a moderate aroma, and 4 indicating a strong aroma. The samples were sniffed by an expert panel and an assessment score was recorded.

### 4.7. Measurement of the Free Amino Acid Contents

The amounts of free amino acids (including Glu, glutamic acid; GABA, gamma aminobutyric acid; Pro, proline) were determined using a previously described methodology [50,81]. A total of 0.1 g of leaves were snapped frozen in liquid nitrogen and ground in a 3% (*w/v*) aqueous solution of 5-sulfosalicylic acid (1-unit acid:9-unit H_2_O). The homogenate was shaken at 200× *g* for 1 h at 37 °C and then centrifuged for 10 min at 12,000× *g*. After passing through a 0.22 mm Millex-LG membrane (Waters, Milford, MA, USA), the supernatant was utilized to analyze the free amino acids (FAAs) using an L-8900 High Speed Amino Acid Analyzer (Hitachi, Tokyo, Japan).

### 4.8. Measurement of Malondialdehyde (MDA)

The malondialdehyde (MDA) content was determined using a malondialdehyde (MDA) test kit (Solarbio, Beijing, China), as previously reported [50,82]. A total of 0.1 g of plant leaf tissues was homogenized in 1 mL of extraction buffer, and the homogenate was centrifuged at 8000× *g* for 10 min at 4 °C. The supernatant was mixed with the working solution, incubated at 100 °C for 60 min, and promptly cooled on ice. After ten minutes of centrifugation at 10,000× *g* at room temperature, the absorbances at 450, 532, and 600 nm were measured, and the MDA content estimated.

### 4.9. Measurement of the Hydrogen Peroxide (H_2_O_2_) Content

Hydrogen peroxide (H_2_O_2_) concentrations were determined using a hydrogen peroxide test kit (Solarbio, Beijing, China), as previously described [83]. Approximately 0.1 g of plant leaf tissues were crushed in liquid nitrogen and homogenized in 1 mL of pre-cooled acetone. After centrifugation at 8000× *g* for 10 min at 4 °C, an aliquot of the supernatant was vortexed with an equivalent amount of hydrogen peroxide detection reagent and incubated for 5 min at room temperature. The absorbance at 415 nm was obtained, and the H_2_O_2_ concentration was evaluated using a standard H_2_O_2_ solution (2 mol/mL).

### 4.10. Antioxidant Enzyme Activity Assays

The activity of the antioxidant enzymes, such as superoxide dismutase (SOD), catalase (CAT), and ascorbate peroxidase (APX), was evaluated in accordance with the methods of the corresponding test kits (Solarbio, Beijing, China) as previously reported [50,84,85]. To determine the antioxidant enzyme activity, approximately 0.1 g of plant leaf was homogenized in 1 mL of extraction buffer, and the homogenate was centrifuged at 8000× *g* for 10 min (20 min for APX) at 4 °C. The working solution for the enzyme activity was produced, and the absorbances at 560 (SOD), 240 (CAT), and 290 nm (APX) were measured, and the enzyme activity was estimated. The unit of the SOD enzyme activity (U/g fresh weight) was defined as the amount of enzyme activity inhibited by 50% in the xanthine oxide coupling reaction system. The activity unit of the CAT enzyme (U/g fresh weight) was defined as the amount of plant tissue in the reaction system that catalyzed the degradation of 1 nmol H_2_O_2_ per minute. The activity unit of the APX enzyme (U/g fresh weight) was defined as 1 mol of ascorbic acid (AsA) oxidized every minute per gram of material.

### 4.11. RNA Extraction and Gene Expression Analysis

The total RNA was extracted from three biological replicates of leaf samples using the RNeasy Plant Mini Kit, as directed by the manufacturer (Qiagen, Hilden, Germany). To determine the concentration and purity of RNA, the NanoDrop 2000 (Thermo Fisher Scientific, Wilmington, DE, USA) was utilized. For the cDNA synthesis, the GoScriptTM Reverse Transcription System (Promega, WI, USA) was utilized, and for the qRT-PCR, the SYBR Green GoTaq^®^ qPCRMaster Mix (Promega, WI, USA) was employed. The rice *OsActin* gene (*LOC_Os08g28190*) was used as an internal control, and the 2^−ΔΔCT^ technique was used to assess the relative expression levels [86]. Appendix A contains a comprehensive list of the qRT-PCR primers.

### 4.12. Statistical Analysis

The mean, standard deviation, and differences between the recorded characters were analyzed for significance using conventional analysis of variance (ANOVA) procedures using the MINITAB-18 program (https://www.minitab.com/en-us/ accessed on 1 November 2020). Tuckey’s test was used to determine the difference between the group mean values at a 5% level of probability.

## 5. Conclusions

In conclusion, a precise knockout mutation of the *OsBadh2* gene resulted in aroma emission but might demonstrate contradictory tolerant phenotypes in mutant lines due to the genotypic differences. A lower mean increment rate of the malondialdehyde, superoxide dismutase, glutamate, and proline content but a higher mean increment rate of γ-aminobutyric acid, hydrogen peroxide, and catalase represented highly tolerant HD mutants. Furthermore, the upregulation of the *OsNAC3* and *OsGAD3* genes in one cultivar (Huaidao#5) but the downregulation in another cultivar (Jiahua#1) exhibited the varietal response differences to salt stress. Thus, the *osbadh2* mutant plants responded to salt stress in a complex way by employing an interacting network of molecular, physiological, and biochemical stress-responsive elements, and the nature of the response was determined by the genetic makeup of the rice variety. More molecular investigations are necessary to explain the molecular basis of the aroma synthesis and salt stress response in rice as well as to explore the possibility of producing highly tolerant aromatic rice genotypes from non-aromatic backgrounds.

## Figures and Tables

**Figure 1 plants-11-02829-f001:**
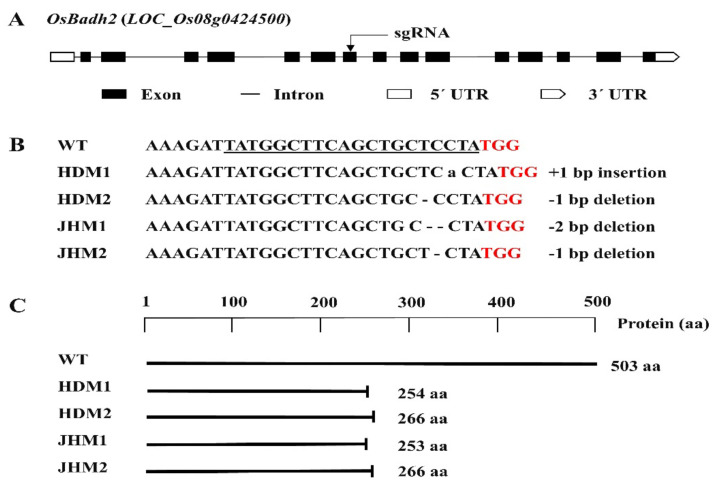
Details of the *OsBadh2* gene-targeted mutagenesis using the CRISPR-Cas9 system: (**A**) diagram of the *OsBadh2* gene’s structure with the target region, exon, intron, and UTRs; (**B**) DNA sequences of the target region with PAM (red), sgRNA (underline), and 1-2 bp insertion or deletion; (**C**) schematic presentation of the protein and truncation in protein translation in the wild type and mutants.

**Figure 2 plants-11-02829-f002:**
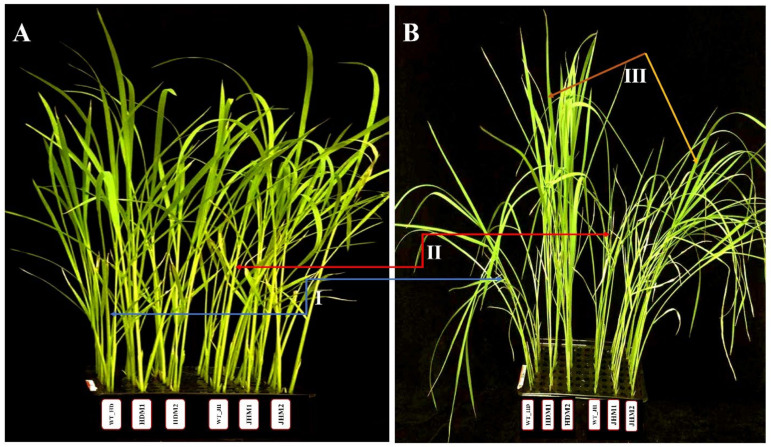
Phenotypic changes in wild-type and mutant lines under salinity treatment: (**A**) control (nontreated); (**B**) 100 mM NaCl treated (after 96 h). WT_HD, Huaidao#5 wild type; HDM1, Huaidao#5 mutant line 1; HDM2, Huaidao#5 mutant line 2; WT_JH, Jiahua#1 wild type; JHM1, Jiahua#1 mutant line 1; JHM2, Jiahua#1 mutant line 2. The blue arrow (I) shows phenotype changes in Huaidao#5, the red arrow (II) shows changes in Jiahua#1, and the brown and yellow arrows (III) show changes among mutant lines.

**Figure 3 plants-11-02829-f003:**
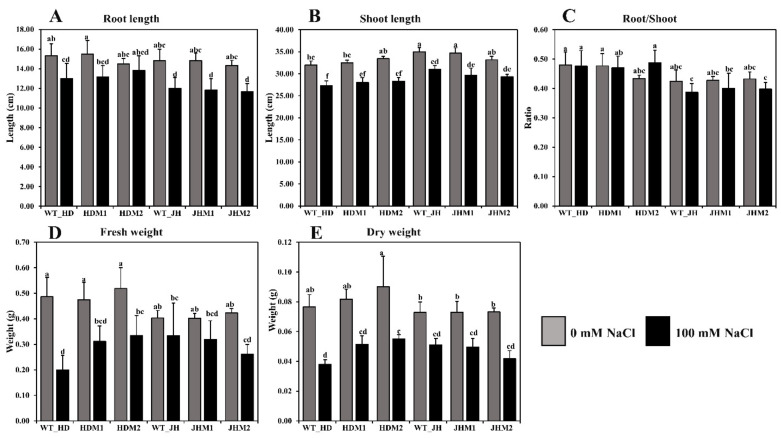
Differences in the morphophysiological parameters under salinity stress: (**A**) root length (cm); (**B**) shoot length (cm); (**C**) root-to-shoot ratio; (**D**) fresh weight (g); (**E**) dry weight (g). Grouping information was obtained using the Tukey method at a 95% confidence level. Means that do not share the same letter are significantly different at a 5% level.

**Figure 4 plants-11-02829-f004:**
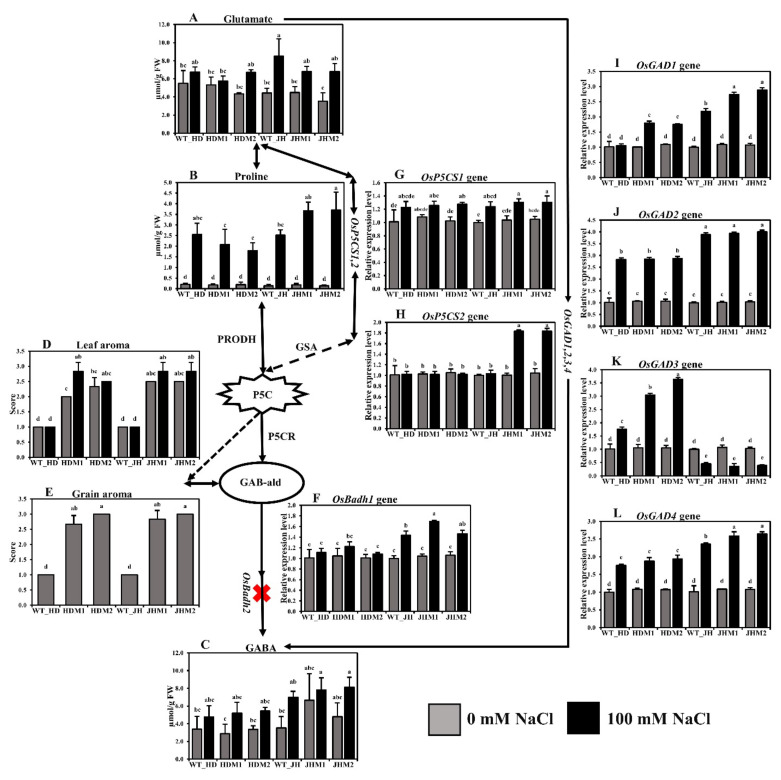
Biosynthesis pathway of rice aroma with gene expression and free amino acid content: (**A**) glutamate content; (**B**) proline content; (**C**) γ-aminobutyric acid (GABA) content; (**D**) leaf aroma; (**E**) grain aroma. The aroma score was calculated based on the 1–4 scale, where 1—no aroma; 2—mild aroma; 3—moderate aroma; 4—excellent aroma; (**F**) relative expression of *OsBadh1*; (**G**) relative expression of *OsP5CS1*; (**H**) relative expression of *OsP5CS2*; (**I**) relative expression of *OsGAD1*; (**J**) relative expression of *OsGAD2*; (**K**) relative expression of *OsGAD3*; (**L**) relative expression of *OsGAD4*. The rice cultivars Huaidao#5 (WT_HD) and Jiahua#1 (WT_JH) were the experimental control (nontreated) for their respective mutant lines, and *OsActin* was an internal reference in the 2^−∆∆CT^ method. Data are shown as the mean and standard deviation of 3 biological replicates with 3 technical replications. Data with different letters (a, b, c, etc.) represent significance at a 0.05 level, using the Tukey Method. PRODH, proline dehydrogenase; GSA, gamma glutamyl semialdehyde; P5C, ∆^1^-pyrroline-5-carboxylate; P5CR, ∆^1^-pyrroline-5-carboxylate reductase; GAB-ald, γ-aminobutyraldehyde; red cross sign (X), knockout mutations in *OsBadh2* gene.

**Figure 5 plants-11-02829-f005:**
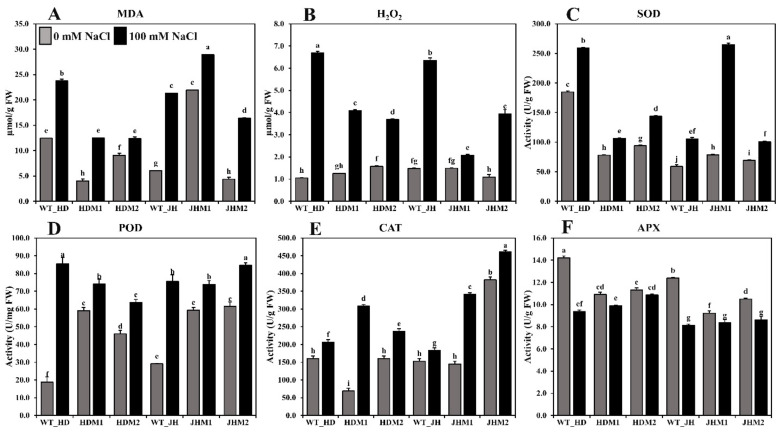
Lipid peroxidation, ROS scavenging, and antioxidant enzyme content in rice seedlings (wild types and mutants). (**A**) MDA content; (**B**) H_2_O_2_ content; (**C**) SOD; (**D**) POD (peroxidase); (**E**) CAT; (**F**) APX enzyme activity measurements. All analyses were performed with three biological replicates and three technical replicates and grouped, using the Tukey method. Means that do not share the same letter are significantly different at a 5% level.

**Figure 6 plants-11-02829-f006:**
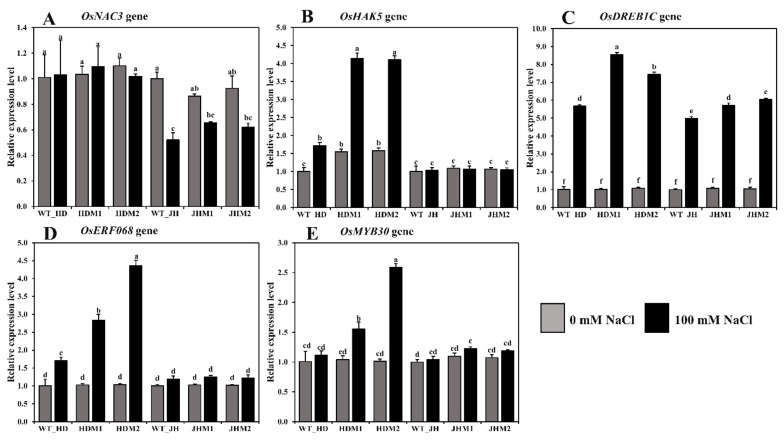
Relative expression of some salinity-related genes in rice. The rice cultivars Huaidao#5 (WT_HD) and Jiahua#1 (WT_JH) were the experimental control (nontreated) for their respective mutant lines, and *OsActin* was an internal reference. (**A**) *OsNAC3*; (**B**) *OsHAK5*; (**C**) *OsDREB1C*; (**D**) *OsERF086*; (**E**) *OsMYB30* gene expression in folds (2^−∆∆CT^) methods. Data are shown as the mean and standard deviation of 3 biological replicates and 3 technical replicates. Data with different letters (a, b, c, etc.) represent significance at a 0.05 level using the Tukey method.

**Figure 7 plants-11-02829-f007:**
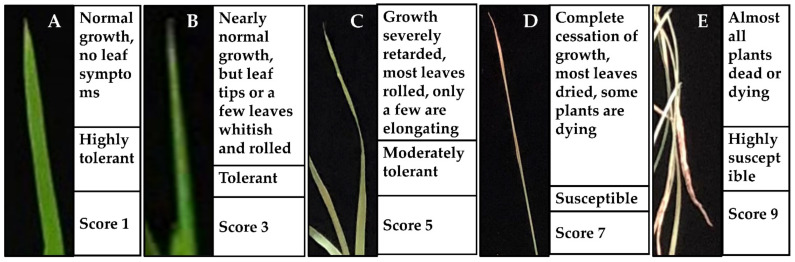
Standard evaluation system (SES) of the salinity response in rice leaves. (**A**) highly tolerant; (**B**) tolerant; (**C**) moderately tolerant; (**D**) susceptible; (**E**) highly susceptible.

## Data Availability

Not applicable.

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
