# Peer review of "Impact of OsBadh2 Mutations on Salt Stress Response in Rice"

_plants, 2022, doi:10.3390/plants11212829_

Round 1

Reviewer 1 Report

The article “Impact of OsBadh2 mutations on salt stress response in rice 2” by Zakaria H. Prodhan,  Shah A. Islam, Shah Alam, Shan Li, Meng Jiang , Yuanyuan Tan ,and Qingyao Shu is a well written manuscript that presented evidence for the complexity of the participation of the Betaine aldehyde dehydrogenase 2 (OsBadh2) gene on  salt stress response.

The introduction contains enough information to establish the relevance of this study because the lack of functional OsBadh2 is highly associated to the aromatic varieties.  The relation between aromatic varieties and salt stress response is also presented.  However, it is important to present more information about the differences between OsBadh1 and OsBadh2 related to function, expression, etc.

The results are clear and concise in general.  In figure 2 is not very easy to visualize the phenotype changes.  It has to have a more specific presentation.

Minor revisions

Line 90 mutations instead of mutantions

Author Response

Response to reviewer 1’s for overall comments:

We would like to express our appreciation to the reviewer for the evaluation and comments. Yes, we have improved the introduction and results sections for a clearer presentation.

Comments and Suggestions for Authors

The article “Impact of OsBadh2 mutations on salt stress response in rice 2” by Zakaria H. Prodhan, Shah A. Islam, Shah Alam, Shan Li, Meng Jiang, Yuanyuan Tan, and Qingyao Shu is a well written manuscript that presented evidence for the complexity of the participation of the Betaine aldehyde dehydrogenase 2 (OsBadh2) gene on salt stress response.

The introduction contains enough information to establish the relevance of this study because the lack of functional OsBadh2 is highly associated to the aromatic varieties. The relation between aromatic varieties and salt stress response is also presented. However, it is important to present more information about the differences between OsBadh1 and OsBadh2 related to function, expression, etc.

1. Response to the reviewer:

It is a great pleasure for us to get such an encouraging comment on our manuscript presentation. We have included more information on the function and expression of OsBadh1 and OsBadh2 related to aroma and salt stress response in the introduction section.

2. Reviewer’s Comments

The results are clear and concise in general. In figure 2 is not very easy to visualize the phenotype changes. It has to have a more specific presentation.

Response to the reviewer:

 Thanks to the reviewer for nicely judging our results. We have improved Figure 2 for a clear visualization of the phenotypic changes.

3. Reviewer’s Comments

Minor revisions

Line 90 mutations instead of mutantions

Response to the reviewer:

 We are thankful to the reviewer for suggesting minor revisions. We have corrected the word “mutantions” to “mutations” in line 90 of our revised manuscript.

Reviewer 2 Report

The review of the article entitled: “Impact of OsBadh2 mutations on salt stress response in rice”.

The authors used the CRISPR-CAS9 method to generate mutations of gene encoding betaine aldehyde dehydrogenase 2 in two non-aromatic rice varieties Huaidao#5 (WT_HD) and Jiahua#1 (WT_JH). The rice lines carrying mutations in the OsBadh2 gene are known to have a stronger aroma however are more susceptible to many environmental stresses including salt stress. The resistance of the obtained aromatic lines to salt stress is investigated.

The authors investigate the phenotype of obtained mutant lines after the exposition to salt stress. The leaf rolling and wilting are observed and based on obtained results authors characterized the HD mutants as improved or highly tolerant and JHM mutants as reduced or moderately tolerant plants. However, the criteria of this evaluation remain not clear. In the “Materials and Method” section the authors refer standard evaluation system but it should be described in more detail. It is also not clear how, in terms of the SES, the salt stress influences the phenotype of the mutants in the reference to WT varieties of rice.

The authors also analysed glutamate, proline, and GABA content, as well as the aroma of leaves and grain and the expression of genes described as “aroma-related”. These results were presented on a thought-out and well-designed diagram, however, some issues related to these results need further clarification. In what way the chosen genes, especially OsGAD1-4 are related to aroma synthesis? What kind of proteins do they encode? What is the GABA shunt pathway and how is it related to aroma synthesis and salinity? In the “Introduction” the role of GABA in response to environmental stresses and aroma synthesis is mentioned only in one sentence. This thread should be significantly expanded to properly introduce the subject of the article.

The changes in lipid peroxidation, hydrogen peroxide accumulation level, as well as antioxidant enzymes activities, were measured. This part of the experiments is well described in terms of results and methodology. My only remark is that there is no expansion of the POD abbreviation in the manuscript text.

The last section of the results contains an expression analysis of the 5 “salinity-related” genes. However why these genes were chosen, what kind of proteins they encode, and what is known about the roles of these proteins in response to salt stress were not described. Such information should be placed in the “Introduction” section.

The “Discussion” section construction is improper since only a small part of it refers directly to the research results and related hypotheses, while large fragments contain general information that should be included in the “Introduction”. What is more, it does not contain any clear conclusions from obtained results. How does the mutation of OsBadh2 influence susceptibility to salt stress? Is it possible to obtain highly aromatic and salt-stress-resistant varieties of rice using this strategy?

These conclusions are also missing in the “Conclusion” section, which is strangely placed after the “Materials and Methods” section instead of after the “Discussion” section.

Also, the “Abstract” needs to be improved since it is not customary to include in the abstract specific numerical data from experiments. It should contain a general description of the main results and conclusions.

In conclusion

The manuscript contains a vast amount of data from well-designed experiments and carries an appropriate dose of scientific novelty. The results are properly and clearly described, however, on some points, they need minor clarifications. Also, the introduction and discussion section, as well as the abstract, should be improved.

Author Response

Response to reviewer 2’s overall comments:

We appreciate the reviewer's assessment and feedback. We have modified the introduction section substantially along with the methods and conclusions parts to make the presentation more understandable.

Comments and Suggestions for Authors

The review of the article entitled: “Impact of OsBadh2 mutations on salt stress response in rice”.

The authors used the CRISPR-CAS9 method to generate mutations of gene encoding betaine aldehyde dehydrogenase 2 in two non-aromatic rice varieties Huaidao#5 (WT_HD) and Jiahua#1 (WT_JH). The rice lines carrying mutations in the OsBadh2 gene are known to have a stronger aroma however are more susceptible to many environmental stresses including salt stress. The resistance of the obtained aromatic lines to salt stress is investigated.

The authors investigate the phenotype of obtained mutant lines after the exposition to salt stress. The leaf rolling and wilting are observed and based on obtained results authors characterized the HD mutants as improved or highly tolerant and JHM mutants as reduced or moderately tolerant plants. However, the criteria of this evaluation remain not clear. In the “Materials and Method” section the authors refer standard evaluation system but it should be described in more detail. It is also not clear how, in terms of the SES, the salt stress influences the phenotype of the mutants in the reference to WT varieties of rice.

1. Response to the reviewer:

We have clarified the criteria of the evaluation system for salt stress response in rice and included Figure 7 in the “Materials and Method” section.

2. Reviewer’s Comments

The authors also analysed glutamate, proline, and GABA content, as well as the aroma of leaves and grain and the expression of genes described as “aroma-related”. These results were presented on a thought-out and well-designed diagram, however, some issues related to these results need further clarification. In what way the chosen genes, especially OsGAD1-4 are related to aroma synthesis? What kind of proteins do they encode? What is the GABA shunt pathway and how is it related to aroma synthesis and salinity? In the “Introduction” the role of GABA in response to environmental stresses and aroma synthesis is mentioned only in one sentence. This thread should be significantly expanded to properly introduce the subject of the article.

Response to the reviewer:

In our modified manuscript, we have included information on the reasons for choosing genes and the relationships of OsGAD genes with aroma and salt stress tolerance. We have also clarified the involvement of the GABA shunt pathway in environmental stress and aroma synthesis.

3. Reviewer’s Comments

The changes in lipid peroxidation, hydrogen peroxide accumulation level, as well as antioxidant enzymes activities, were measured. This part of the experiments is well described in terms of results and methodology. My only remark is that there is no expansion of the POD abbreviation in the manuscript text.

Response to the reviewer:

We have included the elaboration of POD (peroxidase) in the text.

4. Reviewer’s Comments

The last section of the results contains an expression analysis of the 5 “salinity-related” genes. However why these genes were chosen, what kind of proteins they encode, and what is known about the roles of these proteins in response to salt stress were not described. Such information should be placed in the “Introduction” section.

Response to the reviewer:

We have modified the introduction section and included all the information on gene selection, encoding proteins, and the roles of these proteins in response to salt stress. We have also included all the relevant data with citations.

5. Reviewer’s Comments

The “Discussion” section construction is improper since only a small part of it refers directly to the research results and related hypotheses, while large fragments contain general information that should be included in the “Introduction”. What is more, it does not contain any clear conclusions from obtained results. How does the mutation of OsBadh2 influence susceptibility to salt stress? Is it possible to obtain highly aromatic and salt-stress-resistant varieties of rice using this strategy?

Response to the reviewer:

The “Discussion section” has been modified and improved substantially. Some of the information has been moved to the introduction section. The conclusion section is also illuminated and placed at the end of discussion.

6. Reviewer’s Comments

These conclusions are also missing in the “Conclusion” section, which is strangely placed after the “Materials and Methods” section instead of after the “Discussion” section.

Response to the reviewer:

As the “Conclusion” section is required be placed after the “Discussion” section according to the the journal format as instructed in the author’s instruction. So we keep the section after the “Discussion” section.

7. Reviewer’s Comments

Also, the “Abstract” needs to be improved since it is not customary to include in the abstract specific numerical data from experiments. It should contain a general description of the main results and conclusions.

Response to the reviewer:

We have revised the abstract and included a broad overview of the results as well as concluding remarks.

8. Reviewer’s Comments

The manuscript contains a vast amount of data from well-designed experiments and carries an appropriate dose of scientific novelty. The results are properly and clearly described, however, on some points, they need minor clarifications. Also, the introduction and discussion section, as well as the abstract, should be improved.

Response to the reviewer:

We would like to express our gratitude and heartfelt thanks to the reviewer for proper evaluation and critical review of our manuscript. We are delighted to receive such judicious comments and suggestions. We have tried our best to improve our article, including the abstract, introduction, results, discussion, materials and methods, as well as the conclusion sections. We hope this corrected version will be the best version of our manuscript.